# Circulating MicroRNAs and Extracellular Vesicle-Derived MicroRNAs as Predictors of Functional Recovery in Ischemic Stroke Patients: A Systematic Review and Meta-Analysis

**DOI:** 10.3390/ijms24010251

**Published:** 2022-12-23

**Authors:** Codrin-Constantin Burlacu, Daniela Ciobanu, Andrei-Vlad Badulescu, Vlad-Florin Chelaru, Andrei-Otto Mitre, Bogdan Capitanescu, Dirk M. Hermann, Aurel Popa-Wagner

**Affiliations:** 1Chair of Vascular Neurology, Dementia and Ageing, University Hospital Essen, University of Duisburg—Essen, 45147 Essen, Germany; 2Department of Internal Medicine, University of Medicine and Pharmacia Craiova, 200349 Craiova, Romania; 3Faculty of Medicine, Iuliu Hatieganu University of Medicine and Pharmacy, 400349 Cluj-Napoca, Romania; 4Department of Pathophysiology, Iuliu Haţieganu University of Medicine and Pharmacy Cluj-Napoca, Victor Babeş Street, No. 2-4, 400012 Cluj-Napoca, Romania

**Keywords:** microRNAs, extracellular vesicles, disease biomarkers, ischemic stroke, prognosis, stroke recovery

## Abstract

Stroke accounts for the second leading cause of death and a major cause of disability, with limited therapeutic strategy in both the acute and chronic phases. Blood-based biomarkers are intensively researched and widely recognized as useful tools to predict the prognoses of patients confronted with therapeutically limited diseases. We performed a systematic review of the circulating biomarkers in IS patients with prognostic value, with a focus on microRNAs and exosomes as predictive biomarkers of motor and cognitive recovery. We identified 63 studies, totalizing 72 circulating biomarkers with prognostic value in stroke recovery, as follows: 68 miRNAs and exosomal-miRNAs being identified as predictive for motor recovery after stroke, and seven biomarkers being predictive for cognitive recovery. Twelve meta-analyses were performed using effect sizes (random-effects and fixed-effects model). The most significant correlation findings obtained after pooling were with miR-21, miR-29b, miR-125b-5p, miR-126, and miR-335. We identified several miRNAs that were correlated with clinical outcomes of stroke severity and recovery after ischemic stroke, providing predictive information on motor and cognitive recovery. Based on the current state of research, we identified serum miR-9 and neutrophil miR-29b as the most promising biomarkers for in-depth follow-up studies, followed by serum miR-124 and plasma miR-125b.

## 1. Introduction

Stroke affects millions of people, accounting for the second leading cause of death and a major cause of disability [1]. Stroke is a heterogenous disease, and its two forms (ischemic and hemorrhagic) differ in terms of clinical features and disease progression; ischemic stroke (IS) is responsible for 80–85% of cases, while 10–15% are classified as hemorrhagic stroke, which includes subarachnoid hemorrhage and intraparenchymal hemorrhage [2,3].

Thrombolysis with rt-PA (intravenous recombinant tissue plasminogen activator) or thrombectomy are therapeutically restricted to a limited category of patients, i.e., those presenting to thrombolysis centers within 3–4.5 h of stroke onset [4]. 

Dysphasia, hemiparesis and incontinence, painful spasticity related to motor impairment, and post-stroke depression (PSD) are some of the most common reported long-term dysfunctions after stroke [5]. Moreover, unmodifiable factors, such as metabolic status of patients, i.e., hyperlipidemia, hyperglycemia, and comorbidities such as diabetes and hypertension have a high impact on post-stroke recovery [6,7].

About 50–60% of post-stroke patients experience motor dysfunction [8] and the majority of stroke survivors experience delayed recovery of motor and sensory function, or even worsening of neurological function [9]. Therefore, neurorehabilitation strategies have been developed to reduce neurological deficit and promote effective sensorimotor integration in motor and sensory-related impairments [10,11,12,13].

Novel therapeutic approaches were evidenced in experimental stroke models with the aim to improve functional recovery after stroke. However, despite promising pre-clinical results obtained using drugs, cell-based therapies, or mesenchymal stem cell-derived exosomes, the translation from bench to bedside has proven to be challenging [14]. 

Although many studies have focused on neuroprotective and recovery strategies in ischemic stroke, there is limited data on circulating biomarkers with prognostic and diagnostic value. Blood-based biomarkers are intensively researched and widely recognized as useful tools with which to predict the prognoses and survival rates of stroke patients [15]. Thus, the use of accessible biomarkers detected by quantitative polymerase chain reaction (qPCR)-based technics from biofluids could contribute to predicting with a high-accuracy long-term sensorimotor recovery [16]. 

MicroRNAs (miRNAs) emerged as post-transcriptional modulators and activators of signaling pathways in numerous diseases [17,18]. Thus, an altered profile of miRNA expression has been reported in blood samples of stroke patients, with multiple dysregulated miRNAs involved in cerebral ischemic-reperfusion processes, such as excitotoxicity, oxidative stress, neuroinflammation, and neuron cell death processes [19]. 

After ischemic injury, different miRNAs are responsible for communication between brain cells and may provide insights into neuronal repair, vasculogenesis, and neurogenesis [20]. Moreover, miRNAs produced by ischemic brain cells could cross the blood-brain barrier (BBB) packed in extracellular vesicles or exosomes, and exhibit remarkable stability in human biofluids, i.e., plasma or serum. There is hope that their expression profiling may directly reflect disease status and prognosis [21,22,23]. 

Therefore, by quantifying miRNA expression profiles and identifying those associated with unfavorable outcomes we could identify patients at risk for delayed and poor functional recovery and guide rehabilitation strategies. Indeed, compared to protein and metabolic biomarkers, stroke-specific miRNAs show high expression levels immediately after ischemic injury in serum and plasma samples [24]. Most importantly, there is an urgent need to find those blood miRNA biomarkers that correlate with long-term outcomes in the first days after stroke onset. In addition, in animal models of stroke, miRNAs and their carriers such as exosomes and other extracellular vehicles (EVs) play modulatory roles in brain remodeling and repair mechanisms of the neurovascular unit [25,26]. 

This systematic review aims to identify and summarize circulating miRNAs and EV-derived miRNAs that could serve as early-warning markers of long-term outcomes and predictors of physical recovery in stroke patients. To this end, we compared multiple studies reporting blood miRNA expression profiles within serum, plasma, and circulating cells in the early and late stages of stroke recovery. Then, we selected all miRNAs reported in electronic databases with prognostic value on the functional recovery of stroke patients. Moreover, whenever multiple comparable studies analyzed the same miRNA biomarker, we used meta-analytic techniques to obtain a pooled estimate of the correlation between miRNA expression and recovery assessment scores.

## 2. Methods

### 2.1. Search Strategy 

We searched the following electronic databases: Medline PubMed, Scopus, Web of Science, Embase, and Crossref. We used different combinations of the following keyword search terms: (microRNA OR miRNA OR miR OR micro-RNA) AND (NIHSS OR NIH Stroke Scale OR NIH Stroke Score OR National Institutes of Health Stroke Scale) OR (modified Rankin Score OR mRS) OR (MMSE OR Mini-mental state examination) OR (MoCA OR Montreal Cognitive Assessment) AND (Stroke OR acute stroke OR ischemic stroke OR ischemic stroke OR cerebral ischemia OR cerebrovascular accident OR cerebral infarction). 

### 2.2. Inclusion and Exclusion Criteria

For our search, we included the following studies: (1) cohort and case-control studies that evaluate the value of miRNAs in the prognosis of AIS (acute ischemic stroke); (2) CT/MRI-based diagnosis of stroke event in patients ≥ age 18 years; (3) studies retrieved between January 2011 and August 2022, with the primary source of quantitative research in a peer-reviewed journal or thesis published in English or German; (4) AIS diagnosis within 24 h after stroke onset; (5) blood collection time between 24 h after stroke up to 1 year after a stroke event, considering the period of 3–6 months after stroke as the most critical period of time for effective rehabilitation and fast recovery in stroke patients [13]; (6) measure(s) of physical and cognitive outcome using validated motor and cognitive assessment scales (e.g., NIHSS, mRS and/or mBI, MMSE, MoCA); and (7) studies which reported a measure for the prognostic value of said biomarker, e.g., sensitivity, specificity, area under the curve (for dichotomial outcomes), or correlation coefficients between miRNA expression and the results of a scale that assessed physical and cognitive outcome. 

We excluded from the searched studies: (1) animal model studies; (2) studies evaluating non-stroke patients; (3) article not available in English; and (4) conference proceedings. Two Chinese-language studies were included because the English-language abstract provided sufficient information for the purpose of this review, with the exception of quality assessment.

### 2.3. Data Extraction and Quality Assessment 

Two reviewers worked independently to extract the following data from eligible studies: demographic data of stroke cohorts (age/sex), inclusion/exclusion criteria, general patient information (comorbidities and stroke-related risk factors), choice of sample for miRNA quantification, miRNA collection timing, expression levels of circulating miRNAs, miRNA quantification method, disability scale used, disability score timing, and the diagnostic and prognostic performance of studied miRNAs. 

### 2.4. Quality Assessment

We assessed the quality of included studies using the REMARK criteria [27]. This quality assessment tool comprised 8 items originally designed for studies of diagnostic and prognostic tumor-marker studies, evaluating the quality of study design and reporting, as well as the thoroughness of biomarker assessment. Similar systematic reviews aimed at identifying circulating biomarkers with prognostic value in stroke recovery used the REMARK criteria for checking the quality of studies included [7,28]. The items were evaluated as “yes” or “no”, and the percentage of “yes” responses was used as an overall quality score. The quality assessment scores of all the included papers in this review are available in Appendix A.

### 2.5. Statistical Analysis

Data analysis was performed using R 4.2 and package meta (correlation meta-analysis) [29,30]. To ensure meaningful pooling in the meta-analysis of correlation, we converted Pearson’s r and Kendall’s τ into Spearman’s ρ, the most commonly reported statistic in the included articles, based on the table provided by Gilpin for meta-analyses [31]. 

Pooling was performed using fixed-effects and random-effects (DerSimonian-Laird [32]) methods, and heterogeneity was evaluated using the I2 and Cochran’s Q (χ^2^) method [33,34]. Due to the small sample sizes in our paper and the low power of publication bias tests [35,36], we did not check for publication bias.

We did not pool across multiple outcome measures (e.g., correlations between miRNA expression and both NIHSS and mRS scores), following the conclusions of Puhan et al. and the recommendations of the Cochrane Scientific Committee [37,38]. Subsequently, all studies in the correlation meta-analysis employed the NIHSS score.

## 3. Results

Based on the search keywords, we identified 889 records in the following electronic databases: Pubmed, Embase, Scopus, Web of Science, and Crossref. After removing all the duplicates, 632 studies were carried out for screening. Subsequently, we screened the retrieved articles based on the abstract and title, leaving 137 studies assessed for eligibility. After a complete full-text analysis of the remaining studies, 63 studies met our inclusion and exclusion criteria and were included in our review (Figure 1). 

Our search strategy followed the Preferred Reporting Items for Systematic Reviews and Meta-Analyses (PRISMA) criteria [39]. The comprehensive search of the 63 included studies yielded a total of 72 circulating biomarkers with predictive values for stroke severity and recovery in IS patients. Specifically, these biomarkers were 60 non-exosomal circulating miRNA species isolated from blood components (plasma, serum, PBMCs, and other circulating cells) and 12 extracellular vesicle-derived miRNAs isolated from serum and plasma exosomes and endothelial microvesicles (Table 1).

### 3.1. Characteristics of Included Studies

The 63 included papers consisted of 55 case-control studies (IS patients vs. non-stroke controls) and eight cohort studies (which defined two subgroups according to long-term outcomes). These articles evaluated the relationship between circulating miRNA biomarkers and stroke severity and disability, summarizing a total of 5842 AIS, 81 TIA (transient ischemic attack), and 3852 control participants. 

A total of 19 studies recruited fewer than 100 individuals, with two studies including fewer than 50 individuals. Of the 63 included studies, 12 included less than 50 IS patients. The main features of the included studies are provided in Table 1: author and year, country, number of patients recruited, featured miRNAs and miRNA sampling time, and clinical outcome studied. Demographic data of patients recruited in our review is provided in Appendix A.

A total of 22 studies (34.9%) included statistically significant differences in the prevalence of comorbidities between study groups; specifically, 21 studies (33.3%) registered a differing frequency of hypertension, 19 (30.1%) diabetes, and three (4.7%) cardiopathy. A single study (1.5%) reported the Charlson Comorbidity Index, which differed between IS patients and controls. 19 studies (30.1%) reported a differing metabolic profile between stroke patients and controls regarding carbohydrate and lipid metabolism and renal and systemic inflammatory markers. 

A total of 23 studies (36.5%) used the Trial of ORG 10172 in Acute Stroke Treatment (TOAST) classification to evaluate etiologic subtypes of ischemic stroke. 

### 3.2. Methodological Assessment

Following Lai et al. [7], the percentages of “yes” among the eight questions of REMARK criteria [27] were used to evaluate the methodological quality of the included studies; the scores are shown in Appendix A. A prospective design was used in all of the included studies and also all authors used pre-defined clinical outcomes; however, only three studies (4.7%) described any form of blinding. The enrollment period for IS patients was satisfactorily specified in 44 studies (69.8%); almost all studies (98.4%) adequately described the measurement method of the biomarkers. Only two studies (3.1%) performed any form of sample size estimation. A total of 26 studies (41.2%) were deemed to correctly take into account candidate variables, either by including differing baseline characteristics in multivariate models, or by showing no significant differences at baseline.

### 3.3. Collection and Profiling of miRNAs-Based Biomarkers

Blood samples were analyzed from 5842 AIS patients, 81 TIA patients, and 3852 controls, involving 72 circulating biomarkers. Plasma, serum, and various circulating cells were possible sources for miRNA extraction (Table 1). Some studies used microarray assays for miRNA selection, then the expression of which was quantified by using RT-qPCR assays.

Of the 72 miRNA-based biomarkers, 42 miRNAs were analyzed from serum (six from serum extracellular vesicles), 43 were collected from plasma (six from plasma extracellular vesicles), and 11 from circulating cells (four from leukocytes, two from lymphocytes, two from neutrophils, two from peripheral mononuclear cells (PBMCs), and one from white blood cells in general). Two studies employed miRNAs isolated from two different sources, such as PBMCs and serum, and neutrophils and plasma [52,68] (Table 1). miRNAs analysis was carried out at different time points from stroke onset: 59 miRNAs within 24 h; six miRNAs within 24 h after thrombolysis; 17 miRNAs within 72 h; three miRNAs at seven days, four miRNAs at 14 days; two miRNAs at 3 months; and three miRNAs within one year (Table 1). 

### 3.4. Prognostic Tools and Prediction of Physical Recovery in IS Patients 

All included studies used validated clinical scales to estimate the neurological recovery and functional outcome of stroke survivors, scales which have been proven to accurately discriminate between poor- and good-outcome patients [103]. For the purpose of the systematic review, we included studies that assessed stroke severity and long-term disability by using various instruments (see below), and we included studies that estimate cognitive and motor function at different time points.

To assess neurological deficit severity and short-term outcomes in IS patients, the National Institutes of Health Stroke Scale (NIHSS), Glasgow Outcome Scale (GOS), and Glasgow Coma Scale (GCS) were measured in early diagnosed IS patients. For long-term prognosis, studies evaluated motor recovery by using the Modified Rankin Score (mRS) and Barthel Index (BI). The Montreal Cognitive Assessment (MoCA), Hamilton Depression Rating Scale, and Mini-Mental State Examination (MMSE) addressed cognitive outcomes in IS patients. 

Outcomes were measured after stroke at different time points, from hospital admission to a certain period after discharge, ranging from 1 to 3 months up to one year. In 58 studies, neurological deficit severity was evaluated using NIHSS: at 6, 24, 48, and 72 h, 7 days, and 3 months after stroke onset. NIHSS was used as a short-term prognostic scale within 24, 48, and 72 h after stroke onset in 43 studies, and four studies assessed the long-term prognosis of IS patients with the NIHSS score at 7 days [62,84] and 3 months [70,88]. GOS score was measured in two studies at 30 days [49,59], whereas in one study physical outcomes within 24 h were evaluated by GCS [48]. 

mRS and BI were used to evaluate long-term prognosis. mRS score was evaluated in 26 studies at different time points: six studies at hospital admission, five studies within 72 h [52,82,86,87,89], two studies at 7 days [52,84], one study at 12 days [42], two studies at 14 days [53,58], and 10 studies at 3 months [53,61,62,70,84,86,90,92,94,97]. BI was evaluated in two studies at patient discharge [42], on admission, and 7 days after stroke onset [52]. 

MMSE score was used to define cognitive recovery in one study at 14 days after stroke [58], whereas MoCA score assessed cognitive impairments in three studies in the chronic phase of stroke patients (within 1 year) [50,51,55]. To measure depression symptoms among post-stroke patients, three studies evaluated HAMD scores within 2–3 weeks and at 3 months after stroke onset [99,100,101]. 

Of the studies measuring mRS outcomes, eight studies defined a poor outcome as mRS > 2, and two studies as mRS ≥ 2, with the remainder not dichotomizing mRS results into poor or favorable prognoses. Finally, of the studies that used the NIHSS scale and defined dichotomial outcomes, an NIHSS score ≥ 7 was considered a poor prognosis.

### 3.5. Clinical Utility of miRNA-Based Biomarkers in Prediction of Neurological Recovery after Stroke

To further analyze the predictive value of included biomarkers as functional recovery tools, we grouped each study by miRNA and extracted miRNA profiling features, namely miRNA source, miRNA collection timing, disability scale used, disability score timing, and the correlation coefficient and/or sensitivity, specificity, and area under the curve (Appendix A).

The 63 included studies analyzed 72 predictive biomarkers in stroke recovery, comparing miRNA levels between unfavorable/poor and favorable/good outcomes of stroke patients. Of the 72 biomarkers evaluated in the early and late phase of stroke recovery, 68 miRNAs and exosomal-miRNAs were analyzed with respect to motor recovery after stroke, i.e., 65 were related to NIHSS score, nine were related to RS score and three to BI score, and seven for cognitive recovery, i.e., three were related to HAMD score, four were related to MoCA score, and three were related to MMSE score.

Correlation coefficients (Spearman, Kendall, and Pearson) between miRNA expressions and neurological endpoints were analyzed for 68 of the 72 biomarkers, involving in total 5451 IS patients, 81 TIA patients, and 3700 controls. The expression of 10 miRNAs (miR-195, -497, -21, -132, -29b, -24, -409-3p, -135b, -185, and -424) were very strongly correlated with neurological outcome [50,51,58,69,75,80,83,87], whereas the levels of 25 miRNAs had a strong correlation with stroke recovery scores. No statistical significance was obtained for 17 miRNAs. The definitions of very strong (|r| > 0.75), strong (0.75 > |r| > 0.5), moderate (0.5 > |r| > 0.25), and weak (|r| < 0.25) correlation refer to the rule of thumb proposed by Cohen in 1988 [104]. The correlation coefficient between each miRNA and distinct clinical endpoints, appreciating motor and cognitive stroke recovery, is provided in Appendix A.

A total of 13 studies performed ROC analyses to analyze the prognosis of IS patients (neurological deficit severity and post-stroke disability) of the 15 circulating biomarkers. Of the 15 differentially expressed miRNAs, five miRNAs (miR-132, let-7i, miR-140-5p, miR-22, and miR-221-3p) [50,55,99,100,101] were upregulated in patients with cognitive impairments after stroke, and 12 miRNAs (miR-24, miR-29b, miR-210, miR-411-5p, miR-124-3p, miR-125b-5p, miR-192-5p, miR-210, miR-185, miR-424, miR-100-5p, and miR-210) [53,61,62,75,82,87,90,102] were downregulated in poor-prognosis patients with motor deficits.

The highest area under the curve (AUC) value for predicting post-stroke cognitive impairment in patients was reported for miR-132 (0.961 (95% CI: 0.931–0.991)) and miR-411-5p exhibited the highest AUC value (0.900) for detecting neurological motor deficit in stroke patients [53]. Moreover, the lowest AUC value (0.647) was reported for miR-192-5p [61] for assessing motor disability. Three articles each reported having AUC values of at least 0.90, with the remainder reporting less than 0.90 [50,53,101]. miR-132 also exhibited the highest sensitivity (94.9%) and specificity (86.7%) [50]. 

The duration from onset to blood collection varied from the acute phase (from 6, 24, and 72 h up to 14 days) to the chronic phase (i.e., within one year). A total of 63 miRNAs were associated with short-term prognosis of cognitive and motor function. A total of 5 miRNAs were associated with long-term prognosis of motor function, assessed by the GOS [49,59], mRS [84], and NIHSS scores [70,88] at 1–3 months. Moreover, cognitive recovery evaluated by the MMSA, MoCA, and HAMD within 1 year was related to four miRNAs [55,99,100,101]. Most of the studies isolated miRNAs at the same time as the patient’s clinical score assessment, specifically at the time of hospital admission of IS patients. However, in five studies, miRNA quantification was made in both the acute and chronic phases. The following miRNAs have been identified as stroke recovery biomarkers, with prognostic values evaluated at different time points: miR-210 was evaluated at 3 days, 7 days, and 14 days [102]; miR-411-5p was isolated from serum before rt-PA, 24 h after rt-PA, and 3 months (90 days) after stroke onset [53]; miR-497 was assessed from serum at admission and at discharge [79]; miR-135b was isolated from serum within 24 h and at day 14 after stroke onset [83]; and MiR-503 from serum was quantified within 72 h + 3 months [86].

In the acute phase of stroke, longitudinal analysis of blood miRNAs at different time points has been identified in four studies as follows: miR-124 at 24, 48, and 72 h [59]; miR-124-3p at 2, 4, and 6 h [63]; serum miR-134 at 24, 48 and 72 h [92]; and miR-223 from leucocytes at 24, 48, and 72 h [96]. Five studies evaluated miRNA expression profiling [53,54,61,62,64] after thrombolysis therapy, with one study assessing miRNA expression before and after thrombolysis therapy [53]. Nine studies evaluated the relation of 11 miRNAs isolated in the acute phase with a clinical score, evaluated in the chronic period (or at least a few days after miRNA collection). Specifically, the concentration of miR-23b-3p and miR-29b-3p as determined within 24 h was correlated with mRS and BI at discharge (within 12 days) [42]; miR-93 isolated within 6 h was associated with BI at 7 days [52]; miR-124 assessed within 72 h was correlated to GOS at 30 days [59]; miR-125b-5p and miR-206 assessed at 24 h after thrombolysis was correlated with NIHSS score on day 7 [62]. miR-29b isolated within 72 h was correlated with NIHSS at 3 months [70]; miR-128 assessed within 72 days correlated with NIHSS at 7 days and mRS at 3 months [84]; miR-451 assessed within 12 h correlated with NIHSS at 3 months [88]; miR-140-5p assessed within 24 h was associated with HAMD at 3 months [99]; and miR-22 assessed within 7 days was correlated with HAMD at 1 month [100].

### 3.6. Meta-Analysis of Correlations

Of the 63 studies of 72 miRNAs that were obtained through the literature search, the following were identified as being usable in the meta-analysis of correlations (i.e., at least two studies to allow for pooling, fitting the predefined inclusion and exclusion criteria, and having computed a correlation with the NIHSS score): let-7i (two studies), miR-9 (four studies), miR-21 (two studies), miR-29b (four studies), miR-124 (three studies), miR-124-3p (two studies), miR-125a-5p (two studies), miR-125b-5p (four studies), miR-126 (two studies), miR-185 (three studies), miR-195 (three studies), and miR-335 (two studies). In total, 29 studies were included, providing in total 33 data points (since a few studies analyzed more than one miRNA). 

The median REMARK score for the 29 studies (percentage of “yes” responses in the quality assessment questionnaire) was 62.5% (i.e., five “yes” responses out of eight items), indicating moderate quality. Two studies only had English-language abstracts which provided sufficient information for pooling, but not for quality assessment. 

One analyzable study [57] performed the correlation in a sample consisting of both patients (n = 40) and controls (n = 40). Of the 29 included studies, 14 used plasma miRNAs (one from plasma extracellular vesicles), nine extracted miRNA from serum (two from serum extracellular vesicles), and five from circulating cells (two from neutrophils, one from peripheral mononuclear cells, and two from white blood cells in general). Subsequently, meta-analyses with pooling were computed for each miRNA. All suffer from very high heterogeneity (I^2^ > 90% and significant Cochran’s Q test results), with the exception of the four studies identified for miR-125b-5p with moderate heterogeneity (I^2^ = 50% and non-significant Cochran’s Q) and the two studies identified for miR-126 (I^2^ = 0%); however, the studies were performed by the same team, at the same center and during overlapping periods, so we cannot exclude patient overlap. 

The miRNAs for which a significant correlation was obtained after pooling (reporting random-effects results for studies with high heterogeneity and fixed-effects estimates for low-heterogeneity studies) are miR-21 (pooled ρ = −0.52, 95% CI: −0.82–−0.01, random-effects model), miR-29b (pooled ρ = −0.57, 95% CI: −0.74–−0.33), miR-125b-5p (fixed-effects: pooled ρ = 0.32, 95% CI: 0.20–0.43, random-effects: pooled ρ = 0.32, 95% CI: 0.15–0.48), miR-126 (pooled ρ = −0.62, 95% CI: −0.86–−0.14, fixed-effects), and miR-335 (pooled ρ = −0.52, 95% CI: −0.76–−0.16, random-effects). 

Starting from the observation that correlation coefficients differ significantly according to the miRNA source, we repeated the meta-analyses after grouping by miRNA source. In total, we were left with eight miRNAs with at least two studies using a similar sample: miR-9 (serum, three studies), miR-29b (circulating cells, three studies), miR-124 (serum, two studies), miR-125b-5p (plasma, three studies), miR-126 (plasma, two studies), miR-185 (plasma, two studies), miR-195 (plasma, two studies), and miR-335 (plasma, two studies). 

After subsetting for sample type, the following significant pooled effects were obtained: serum miR-9 (pooled ρ = 0.68, 95% CI: 0.59–0.74, fixed-effects model), white blood cell miR-29b (pooled ρ = −0.46, 95% CI: −0.57–−0.33, fixed-effects), serum miR-124 (pooled ρ = 0.54, 95% CI: 0.18–0.78, random-effects), plasma miR-125b-5p (pooled ρ = 0.35, 95% CI: 0.23–0.46, fixed-effects), plasma miR-185 (pooled ρ = 0.15, 95% CI: 0.02–0.27, fixed-effects), and plasma miR-195 (pooled ρ = −0.25, 95% CI: −0.46–−0.01). 

Heterogeneity is noticeably lower for miR-9 (from an I2 value of 98% to 0%), miR-29b (from 89% to 0%), and miR-185 (from 96% to 0%) and improved slightly for miR-124 (from 96% to 85%), miR-125b-5p (from 50% to 42%), and miR-195 (from 98% to 75%). 

The resulting forest plots are presented in Figure 2 and Figure 3 comparatively, with both fixed (common)-effects and random-effects estimates and heterogeneity measures.

## 4. Discussion

Currently, the use of miRNAs as prognostic and diagnostic markers has been gaining recognition in stroke research due to their satisfactory stability in peripheral biofluids, and ease of quantification and validation [21,22,23]. Nonetheless, the field of circulating miRNAs as biomarkers in ischemic stroke is still in its early stages, with few studies being performed on the same miRNA and in similar conditions [15].

By conducting a systematic literature search, we identified all eligible studies which address miRNA expression profiles in blood samples of IS patients related to stroke severity and recovery. Our analysis yielded 72 circulating miRNA-based biomarkers, including 60 non-exosomal miRNAs and 12 exosome-derived miRNAs with predictive values for neurological deficit severity, disability, and stroke recovery (Table 1).

To our knowledge, only one systematic review analyzed the diagnostic and prognostic performance of miRNAs in stroke patients [24]. The authors have a different approach to the topic compared to ours, analyzing dichotomous outcomes of dysregulated miRNAs (i.e., sensitivity and specificity in predicting good or poor outcomes) within the acute phase (<24 h) of IS and ICH [24].

Our systematic review addressed for the first time the prognostic performance of circulating miRNA related to IS recovery, comparing correlation coefficients and ROC analysis between differential-expressed miRNAs and neurological assessment scores. For the meta-analysis of correlation coefficients, we included 29 studies, providing in total 33 data points, including only miRNAs retrieved in at least two studies, which were correlated with NIHSS score.

The prognostic value of miRNAs was analyzed in different blood specimens and at different time points for both miRNA quantification and stroke outcomes, offering a comprehensive picture of miRNAs in IS recovery. Of the 72 predictive biomarkers analyzed from ischemic stroke samples, 68 miRNAs exhibited prognostic value for motor recovery, and seven biomarkers indicate prognostic value for cognitive recovery.

The utility of these circulating biomarkers in predicting long-term outcomes of stroke patients comes with some concerns, which mainly stem from the heterogeneity of miRNA expression. The circulating biomarkers identified by our systematic review encounter a large variability in miRNA isolation profile: multiple different time points of blood sample collection; different miRNA sources evaluated; different vascular and metabolic factors harbored by some stroke patients; and different stroke subtypes between stroke patients.

The heterogeneity of miRNA expression is explained in part by the different blood components used for miRNA analysis [105], which is discussed in detail below. In our case, the literature review identified 42 miRNAs analyzed from serum, 43 miRNAs collected from plasma, and 11 miRNAs isolated from circulating cells. A systematic review analyzing predictive biomarkers for physical recovery highlighted the importance of a short window (24 ± 6 h) for blood sampling to provide prognostic information in the early phase of IS [7]. However, miRNA isolated in the acute stroke phase might provide ischemic injury changes rather than stroke recovery changes, according to Edwardson et al. [26]. We included a large window of blood collection timing for miRNA quantification, ranging from immediately after hospital admission (<6, 24, and 72 h) to the chronic phase of stroke (within one year), capturing all miRNAs which exhibited short-term and long-term prognostic value in IS patients.

Nonetheless, our results highlight the importance of more comprehensive blood sampling, ideally through longitudinal, repeated-measures approaches, which would capture the prognostic signature of miRNAs related to stroke recovery, a point also made by Edwardson et al. [26]. However, we identified only four studies that performed a longitudinal analysis of blood miRNAs at different time points: miR-124 [59]; miR-124-3p [63]; miR-134 [92]; and miR-223 [96]. Of particular note, we identified nine studies that evaluated the relationship between miRNAs assessed in the acute phase of stroke (in total 11 miRNAs) with long-term clinical outcomes (i.e., assessed in the chronic phase or at least a few days after miRNA collection) [15].

Another contributor to the heterogeneity of both miRNA expression and stroke outcome is the presence of comorbid conditions: although several studies analyze differential miRNA expression according to stroke subtype or comorbidities [94,106,107], we also identified 22 studies that have statistically significant differences in the prevalence of comorbidities between stroke patients and control groups, including hypertension, diabetes, cardiopathy, metabolic profiles of carbohydrate, lipid metabolism, and renal and systemic inflammatory markers, but do not attempt to investigate whether they act as confounders.

Only 23 (36.5%) studies distinguish between the etiologic subtypes of IS (as defined by the TOAST classification). Of these studies, it is only Tan et al. who investigated differences between miRNA profiles according to TOAST subtype: small-artery (SA), large-artery (LA), and cardioembolic; more specifically, differential miRNA expression between good and poor outcomes is only apparent for SA or LA stroke [108].

Yet another source of variability is heterogeneity in treatment, more specifically between patients undergoing or not undergoing thrombolysis: we identified five studies that evaluated miRNA expression profiling after thrombolysis [53,54,61,62,64]. Nonetheless, Lin et al., evaluated miR-411-5p before and after thrombolysis therapy [53], suggesting that it might have a smaller contribution to heterogeneity. So far, there were no large studies that demonstrated that tissue plasminogen activator might modify miRNA expression profiles, which further impacts the assessment of prognostic biomarkers [54].

Our analysis highlights 10 miRNAs (miR-195, -497, -21, -132, -29b, -24, -409-3p, -135b, -185, and -424) to be very strongly correlated with neurological outcome [50,51,58,69,75,80,83,87]. Of the studies with binary outcomes, miR-132 and miR-411-5p exhibited the highest AUC value (0.961 and 0.900, respectively) for detecting neurological motor deficit in stroke patients [53], with miR-132 exhibiting the highest sensitivity (94.9%) and specificity (86.7%) [50].

The only meta-analysis performed on the topic of circulating miRNAs in stroke patients is that of Deng et al., who investigated the utility of miRNAs in diagnosing ischemic stroke. The authors pooled results from multiple miRNAs [109]. While this approach is shared by a number of other miRNA-focused meta-analyses, we do not consider the pooled coefficients to have an obvious clinical significance. By grouping results by miRNA and then by miRNA source, we intended to obtain more meaningful results, with a higher clinical utility.

The meta-analysis focused on the connection between miRNAs and a quantitative estimate of stroke severity (in our case, NIHSS scores, typically at or soon after diagnosis). A vast majority of the included studies have approached the former topic, with most articles reporting correlation coefficients between miRNA expression and the NIHSS scores, an outcome measure that can easily be pooled.

One of the first observations regarding the meta-analysis of correlations is that the results depend significantly on the choice of miRNA source: serum, plasma, and specific blood cells. The existence of important differences between the miRNAome of plasma and serum in healthy individuals has been known since 2012: a striking observation is that the coagulation process releases cellular miRNAs into serum, which would imply that the miRNome of serum is more similar to that of whole blood than to that of plasma. Additionally, the authors showed that different qPCR platforms are in low agreement. These results are supported by evidence of variable changes in the miRNA profiles of whole blood, serum, and plasma induced by physical disturbance, long-term cold storage, and freeze-thaw cycles [110,111].

A study on patients with non-ST-elevation myocardial infarction has likewise identified several inconsistencies between serum and plasma: the levels of miR-1 and miR-208 are more variable in plasma, the expression of miR-133a and miR-26a are significantly different from controls only in serum, and miR-21 is increased in serum and decreased in plasma compared to controls [105].

Our meta-analysis cannot prove that the choice of serum, plasma, or cells is responsible for the identified heterogeneity, especially since other factors vary between studies (qPCR protocols, timing, patient selection, etc.) and few of the included articles rank well on the REMARK bias assessment. Nonetheless, these results highlight the importance of a comparative primary study of the serum, plasma, and circulating cell miRNA landscapes of stroke patients, in order to identify the most promising biomarkers.

Our meta-analysis lends support to the potential clinical utility and highlights the need for follow-up studies for the following biomarkers: serum miR-9, neutrophil/WBC miR-29b, plasma miR-125b-5p, and plasma miR-195. More specifically, we have found that serum miR-9 shows high performance as an indicator of stroke severity: the pooled Spearman’s correlation coefficient is relatively high (0.68), and there is a low heterogeneity between the three included studies, despite differing techniques (one study used exosomal miR-9), miRNA collection timing, comorbidities (one study included only diabetics with stroke), moderately low-quality assessment scores, and ethnicity (two Chinese studies and one Egyptian study). A slightly weaker association was found for white blood cell miR-29b (ρ = −0.46), but, like serum miR-9, it shows very low heterogeneity across the three included studies.

Plasma miR-125b-5p also shows potential, with a moderate pooled coefficient (0.43) and moderate heterogeneity, and to a lesser extent plasma miR-195 (with a relatively low pooled ρ = −0.25 and moderately high heterogeneity).

The main limitations of our study are represented by all aforementioned sources of heterogeneity in the differential expression of miRNAs, from methods, normalization, and timing of miRNA isolation, and the blood component used for miRNA analysis, to the intrinsic characteristics of the evaluated population. Indeed, different normalization strategies may lead to different outputs [112]. Most of the studies included in our review originate from China, with several studies from Thailand, Germany, Lebanon, the USA, Egypt, and Iran.

Thus, the detection of differential expression of miRNA profiles might vary among the study population. The limitations of our meta-analytic methods stem from the very limited sample sizes: no miRNA has been investigated by more than four studies, a number that drops to two to three after restricting the analysis according to biofluid.

For this reason, we did not test for publication bias, in order to avoid misleading false-negative results due to their low statistical power [35,36]. Nonetheless, very few included studies reported non-significant results, so the pooling results are likely to be skewed due to publication bias.

## 5. Conclusions

In this review, we identified 72 circulating predictive biomarkers of stroke severity and stroke recovery with respect to the motor and cognitive functions of IS patients.

Based on the current state of research, we identified serum miR-9 and neutrophil miR-29b as the most promising biomarkers for in-depth follow-up studies, followed by serum miR-124 and plasma miR-125b. Although circulating miRNAs could contribute to facilitating stroke diagnosis and predicting long-term outcomes, we identified multiple shortcomings which need to be addressed before the inclusion of miRNA-based panels in clinical practice: the lack of studies investigating long-term outcomes, the uninvestigated discrepancy between plasma and serum miRNA expression, and the high risk of bias and low sample size.

## Figures and Tables

**Figure 1 ijms-24-00251-f001:**
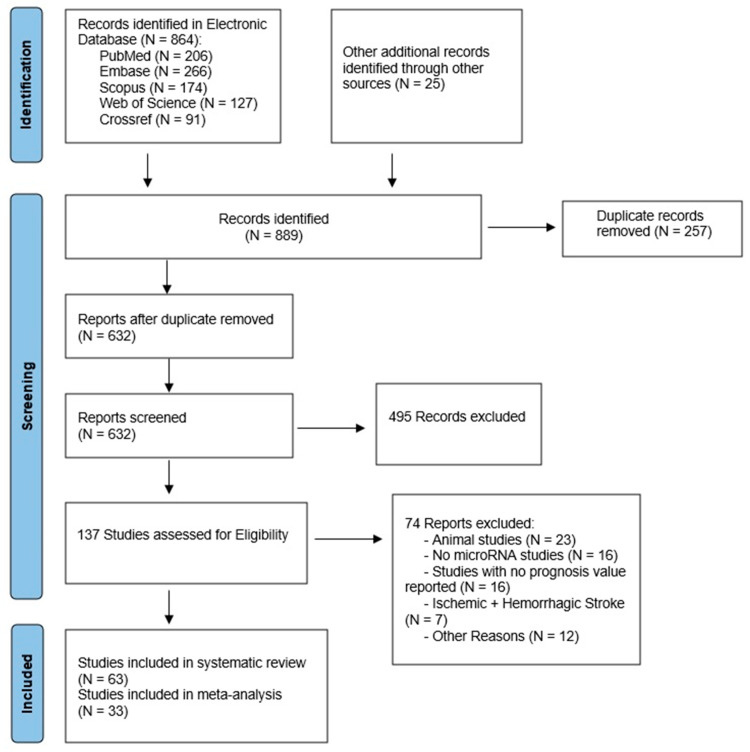
Flowchart with selection criteria of research studies included in review.

**Figure 2 ijms-24-00251-f002:**
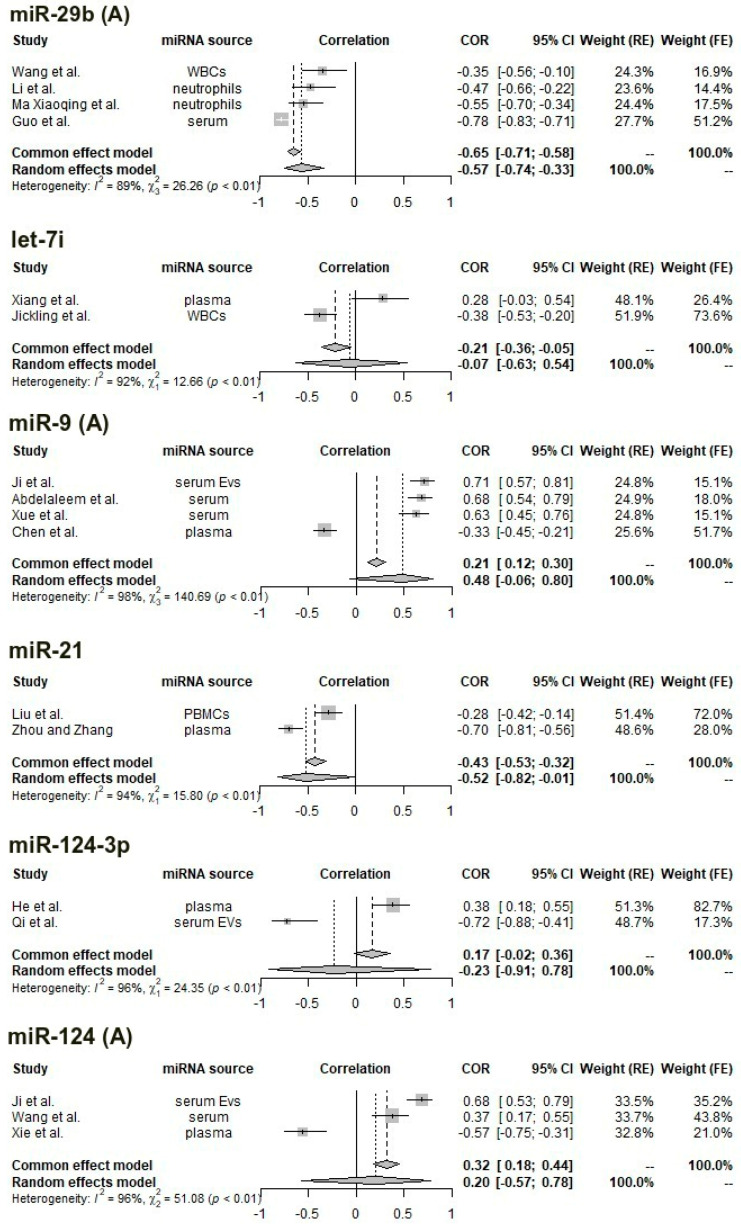
Forest plot for the correlation between the concentration of let-7i, miR-9, miR-21, miR-29b, miR-124, miR-124-3p, and NIHSS score; in certain cases, we address separately all reported studies (**A**) and (**B**) only studies with reported same miRNA source. References: miR-29b: [47,70,71,75]; let-7i: [54,56]; miR-9: [60,66,72,73]; miR-21: [68,69]; miR-124-3p: [61,63]; miR-124: [57,60,74].

**Figure 3 ijms-24-00251-f003:**
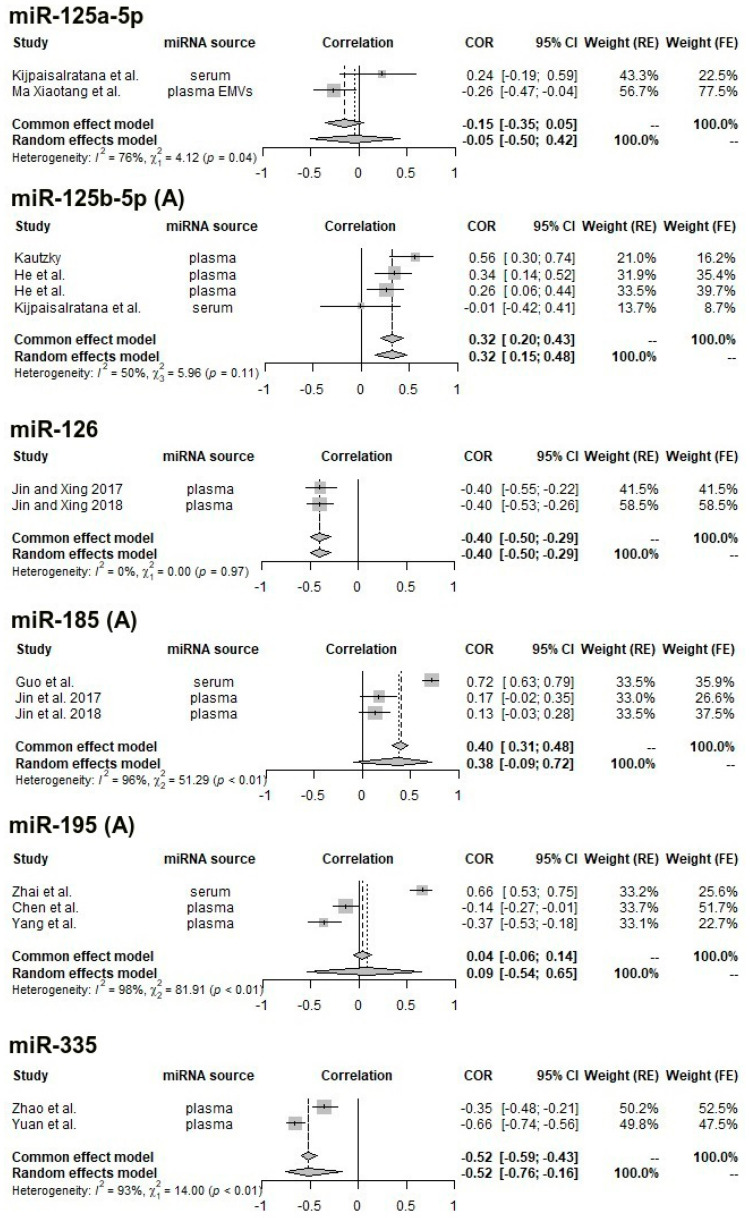
Forest plot for the correlation between the concentration of miR-125a-5p, miR-125b-5p, miR-126, miR-185, miR-195, miR-335, and NIHSS score; in certain cases, we address separately all reported studies (**A**) and (**B**) only studies with reported same miRNA source. References: miR-125a-5p: [40,43]; miR-125b-5p: [43,44,61,62]; miR-126: [64,65]; miR-185: [64,65,87]; miR-195: [51,66,67]; miR-335: [76,78].

**Table 1 ijms-24-00251-t001:** Overview of identified circulating miRNAs in blood specimens which provide prognostic value in stroke patients.

Study and Author	Country	N	Featured miRNAs	MiRNA Source	miRNA Sample Timing	Clinical Outcome Measured
Ma et al., 2022 [40]	China	72/56	EMV-miR-125a-5p	Plasma	<24 h	NIHSS
Peng et al., 2015 [41]	China	72/52	miR-let-7e	Serum	<24 h	NIHSS
Wu et al., 2017 [42]	China	131/50	miR-23b-3p, -29b-3p, -181a-5p, -21-5p	Serum	<24 h	NIHSS, BI, mRS
Kijpaisalratana et al., 2020 [43]	Thailand	23/35	miR-125a-5p, -125b-5p, -433-5p	Serum	<72 h	NIHSS
Kautzky et al., 2022 [44]	Germany	40/40	miR-125a-5p, miR-125b-5p	Plasma	<24 h	NIHSS
Zhu et al., 2019 [45]	China	170/170	miR-143	PBMC	< 24 h	NIHSS
Niu et al., 2021 [46]	China	453/143	Exosomal-miR-369-3p, -493-3p, -379-5p, -1296-5p	Plasma	<72 h	NIHSS
Li et al., 2022 [47]	China	50/42	miR-29b	Leukocytes	<6 h	NIHSS
Kotb et al., 2019 [48]	Lebanon	44/22	miR-146a	Serum	<24 h	GCS
Wu et al., 2020 [49]	China	112/112	miR-99b	Plasma	NR	GOS
Huang et al., 2016 [50]	China	76/38	miR-132	Serum	Chronic	MoCA
Zhai et al., [51]	China	108/76	miR-195, -497	Serum	<72 h	NIHSS, MoCA
Ma et al., 2019 [52]	China	33/20	miR-93	Neutrophils and plasma	<6 h	NIHSS, BI, mRS
Lin et al., 2022 [53]	China	96	miR-411-5p	Serum	Before rt-PA, 24 h after rt-PA, and at 3 months	NIHSS
Xiang et al., 2017 [54]	China	40/46	let-7i	Plasma	24 h after thrombolysis	NIHSS
Wang et al., 2020 [55]	China	76	let-7i	Serum	Chronic (within 1 year)	MoCA
Jickling et al., 2016 [56]	USA	106/106	let-7i	Circulating leukocytes	<72 h	NIHSS
Wang et al., 2019 [57]	China	40/40	miR-124	Serum	<24 h	NIHSS
Yuan et al., 2022 [58]	China	77	miR-21, -132, -200b	Serum	14 days	MMSE
Zhou and Qi, 2021 [59]	China	108/108	miR-124	Serum	At 24, 48, and 72 h	GOS
Ji et al., 2016 [60]	China	65/66	Exosomal-miR-9, -124	Serum	Mean 16.5 h	NIHSS
He et al., 2019 [61]	China	94	miR-124-3p, -125b-5p, -192-5p, -125b-5p	Plasma	24 h after thrombolysis	NIHSS
He et al., 2019 [62]	China	84	miR-125b-5p, -206	Plasma	24 h after thrombolysis	NIHSS
Qi et al., 2021 [63]	China	10/10	Exosomal-miR-124-3p	Serum	At 2, 4, 6 h	NIHSS
Jin and Xing, 2017 [64]	China	106/110	miR-126, -130a, -185, -221, -222, -218, -378, -19a, -296, -101, -206	Plasma	<24 h (after thrombolysis)	NIHSS
Jin and Xing, 2018 [65]	China	148/148	miR-126, -218, -130a, -185, -222	Plasma	<24 h	NIHSS
Chen et al., 2020 [66]	China	215/215	miR-9, -195	Plasma	<24 h	NIHSS
Yang et al., 2018 [67]	China	96/45	miR-195	Plasma	<72 h	NIHSS
Liu et al., 2021 [68]	China	170/100	miR-21	PBMCs + serum	<24 h	NIHSS
Zhou and Zhang, 2014 [69]	China	68/21	miR-21	Plasma	<24 h	NIHSS
Wang et al., 2015 [70]	China	58/59	miR-29b	WBCs	<72 h	NIHSS
Ma et al., 2020 [71]	China	60/40	miR-29b	Neutrophils	<6 h	NIHSS
Abdelaleem et al., 2022 [72]	Egypt	77/71	miR-9, -106a	Serum	At diagnosis	NIHSS
Xue et al., 2018 [73]	China	65/55	miR-9	Serum	At diagnosis	NIHSS
Xie et al., 2019 [74]	China	40 IS + 40 Controls	miR-124	Plasma	NR	NIHSS
Guo et al., 2020 [75]	China	170/65	miR-24, -29b	Serum	NR	NIHSS
Yuan et al., 2016 [76]	China	152/136	miR-335	Plasma	<24 h	NIHSS
Yuan et al., 2016 [77]	China	152/136	miR-26b	Plasma	<24 h	NIHSS
Zhaou et al., 2016 [78]	China	168/104	miR-335	Plasma	<24 h	NIHSS
Zhong et al., 2021 [79]	China	89/39	miR-497	Serum	At admission (<24 h) and at discharge	NIHSS
Song et al., 2021 [80]	China	80/30	miR-409-3p	Serum	<9 h	NIHSS
Zhang et al., 2020 [81]	China	93/70	EMVs-miR-155	Plasma	(<24 h)	NIHSS
Zeng et al., 2013 [82]	China	105	miR-210	Leukocytes	<72 h	NIHSS, mRS
Yang et al., 2020 [83]	China	79/60	miR-135b	Serum	<24 h and 14 days after admission	NIHSS
Liu et al., 2019 [84]	China	40/25	miR-128	Lymphocytes	<72 h	NIHSS, mRS
Zhao et al., 2020 [85]	China	43/26	miR-494	Lymphocytes	<6 h	
Sheikhbahaei et al., 2019 [86]	Iran	33/17	miR-503	Serum	<72 h + 3 months	NIHSS, mRS
Guo et al., 2022 [87]	China	142/50	miR-185, -424	Serum	<24 h	NIHSS, mRS
Fu et al., 2019 [88]	China	108/97	miR-451	Whole blood	<12 h	NIHSS
Ye et al., 2021 [89]	China	43/43	Exosomal-miR-27-3p	Serum	On admission	NIHSS
Otero-Ortega et al., 2021 [90]	Spain	81/22	miR-100-5p	Serum	<24h	NIHSS, mRS
Yang et al., 2016 [91]	China	114/58	miR-107, -128b, -153	Plasma	<24 h	NIHSS
Zhou et al., 2018 [92]	China	50/50	Exosomal-miR-134	Serum	24-, 48- and 72 h	NIHSS
Chen et al., 2018 [93]	China	128/102	miR-146b	Serum	< 24 h	NIHSS
Chen et al., 2017 [94]	China	50/33	Exosomal-miR-223	Serum	< 72 h	NIHSS
Liang et al., 2016 [95]	China	102/97	miR-34a-5p	Plasma	<12 h	NIHSS
Wang et al., 2014 [96]	China	79/75	miR-223	Leucocytes	At 24, 48, 72 h	NIHSS
Wang et al., 2021 [97]	China	88/88	miR-9-5p, -128-3p	Serum	<6 h	mRS
Jia et al., 2015 [98]	China	146/96	miR-145	Serum	<24 h	NIHSS
Liang et al., 2019 [99]	China	62/62	miR-140-5p	Plasma	<24 h	HAMD
Hu et al., 2020 [100]	China	257	miR-22	Plasma	<7 days	HAMD
Cui et al., 2021 [101]	China	136	miR-221-3p	Serum	Chronic	NIHSS, HAMD
Zeng et al., 2011 [102]	China	112/60	miR-210	Leucocytes	<3 days, 7 days, 14 days	NIHSS

Legend: IS = ischemic stroke; N = number of IS patients/controls; Abbreviations: BI, Barthel Index; EMV, endothelial microvesicles; GCS, Glasgow Coma Scale; GOS, Glasgow Outcome Scale; HAMD, Hamilton Depression Rating Scale; MoCA, Montreal Cognitive Assessment; MMSE, Mini-Mental State Examination; mRS, Modified Rankin Score; NIHSS, National Institutes of Health Stroke Scale; PBMC, peripheral blood mononuclear cells.

## Data Availability

Upon reasonable request from the corresponding author.

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
