# Peer review of "Circulating MicroRNAs and Extracellular Vesicle-Derived MicroRNAs as Predictors of Functional Recovery in Ischemic Stroke Patients: A Systematic Review and Meta-Analysis"

_ijms, 2022, doi:10.3390/ijms24010251_

Round 1

Reviewer 1 Report

This systematic review and meta-analysis  assessed whether blood biomarkers in patients with ischemic stroke have prognosis value. The study focuses on microRNAs and microRNAs in exosomes as predictive biomarkers of motor and cognitive recovery. The article is solid because it particularly mentions the heterogeneity with regard to the methods of microRNA isolation and detection. I have the following comments

1. Pleasen shorten the Introduction.
2. Page 3, line 113: please elaborate how differences in normalization of microRNA expression data are taken into account in your inclusion of studies.
3. Page 5, line 183: you include 81 TIA patients. Why? Explain the relevance of those patients since a TIA diagnosis is often uncertain.

Author Response

This systematic review and meta-analysis  assessed whether blood biomarkers in patients with ischemic stroke have prognosis value. The study focuses on microRNAs and microRNAs in exosomes as predictive biomarkers of motor and cognitive recovery. The article is solid because it particularly mentions the heterogeneity with regard to the methods of microRNA isolation and detection. I have the following comments

  1. Please shorten the Introduction.

Answer: The Reviewer is right. The Introduction was a bit unfocused and excessively long. In the revised version, INTRODUCTION has been cut by some 20% and text edited to convey the main message of the article.

  1. Page 3, line 113: please elaborate how differences in normalization of microRNA expression data are taken into account in your inclusion of studies.

Answer: The Reviewer is right, different normalization strategies may lead to different outputs (Faraldi et al., 2019). This is another study limitation that has been added along with the reference to the manuscript.

  1. Page 5, line 183: you include 81 TIA patients. Why? Explain the relevance of those patients since a TIA diagnosis is often uncertain.

Answer: The reviewer is certainly right. However, we find of interest to clinicians to include patients with TIA because miRNA biomarkers would be in this case particularly useful to identify risk factors and start appropriate medication

Reviewer 2 Report

In the review untitled Circulating microRNAs and Extracellular Vesicles-derived microRNAs as Predictors of Functional Recovery in Ischemic Stroke Patients: A Systematic Review and Meta-analysis”, Codrin-Constantin and colleagues present results of a systematic review and meta-analysis aimed to identify circulating biomarkers (miRNAs and exosomal-miRNAs) having prognostic value for ischemic stroke severity and stroke recovery. Based on the analyzed publications, the Authors identified several miRNAs that correlated with clinical outcomes of stroke severity and miRNAs  that correlated with ischemic stroke recovery. Authors indicate miR-9 and miR-124 (in serum), miR-29b (from neutrophils), and  miR-125b (in plasma) as the most promising biomarkers that should be further investigated. I find the subject of the review very interesting and clinically relevant. A tremendous amount of work has been done and very detailed analysis presented. The applied methodology is suitable.

 Below you find my comments.

1. The Introduction is a bit unfocused – paragraphs stand alone and the text flow could be better (no “funnel” structure i.e. from general to more detailed presentation of the scientific problem). e.g.: lines 34-38 why to provide information about hemorrhagic stroke while the analysis concerns IS patients? I suggest to re-write this section.

2. lines 67-69 – the reference supporting the statement that “blood-based biomarkers are intensively researched and widely recognized as useful tools to predict prognosis and survival rate of patients (…)” is related to study involving patients with Alzheimer’s disease. Why not focus on ischemic stroke itself and successful/unsuccessful attempts of looking for biomarkers of diagnostic and prognostic value. Such a summary is presented in review by Steliga and colleagues (PMID: 31701356). Authors should consider citing this article.

3. In some sentences, unscientific/colloquial language has been used. A few examples:

Lines 409-411: “(…) field of circulating miRNAs as biomarkers of ischemic stroke is still in its infancy (…)”

Lines 458-460: “Of particular note, we identified (…)”

These sentences shouors should consider citing this article.ld be rewritten, and the entire text checked for non-scientific/colloquial content and corrected.

Author Response

: In the review untitled „Circulating microRNAs and Extracellular Vesicles-derived microRNAs as Predictors of Functional Recovery in Ischemic Stroke Patients: A Systematic Review and Meta-analysis”, Codrin-Constantin and colleagues present results of a systematic review and meta-analysis aimed to identify circulating biomarkers (miRNAs and exosomal-miRNAs) having prognostic value for ischemic stroke severity and stroke recovery. Based on the analyzed publications, the Authors identified several miRNAs that correlated with clinical outcomes of stroke severity and miRNAs  that correlated with ischemic stroke recovery. Authors indicate miR-9 and miR-124 (in serum), miR-29b (from neutrophils), and  miR-125b (in plasma) as the most promising biomarkers that should be further investigated. I find the subject of the review very interesting and clinically relevant. A tremendous amount of work has been done and very detailed analysis presented. The applied methodology is suitable.

 Below you find my comments.

  1. The Introduction is a bit unfocused – paragraphs stand alone and the text flow could be better (no “funnel” structure i.e. from general to more detailed presentation of the scientific problem). e.g.: lines 34-38 why to provide information about hemorrhagic stroke while the analysis concerns IS patients? I suggest to re-write this section.

Answer: The Reviewer is right. The Introduction was a bit unfocused and excessively long. In the revised version the Introduction has been shortened and re-arranged to better convey the scientific message.

  1. lines 67-69 – the reference supporting the statement that “blood-based biomarkers are intensively researched and widely recognized as useful tools to predict prognosis and survival rate of patients (…)” is related to study involving patients with Alzheimer’s disease.Why not focus on ischemic stroke itself and successful/unsuccessful attempts of looking for biomarkers of diagnostic and prognostic value. Such a summary is presented in review by Steliga and colleagues (PMID: 31701356). Authors should consider citing this article.

Answer: Thank you for useful comments. The reference has been added at the indicated locations.

  1. In some sentences, unscientific/colloquial language has been used. A few examples:

Lines 409-411: “(…) field of circulating miRNAs as biomarkers of ischemic stroke is still in its infancy (…)”

Answer: Thank you for useful comments. Done.

Lines 458-460: “Of particular note, we identified (…)”

These sentences should consider citing this article.ld be rewritten, and the entire text checked for non-scientific/colloquial content and corrected.

Answer: Thank you for useful comments. The reference has been added at the indicated locations. English has been checked.